# A Scoping Review of Marginal and Internal Fit Accuracy of Lithium Disilicate Restorations

**DOI:** 10.3390/dj10120236

**Published:** 2022-12-12

**Authors:** Tanya Patel, Neil Nathwani, Peter Fine, Albert Leung

**Affiliations:** UCL Eastman Dental Institute, Rockefeller Building, 21 University Street, London WC1E 6ED, UK

**Keywords:** lithium disilicate, marginal fit, digital impression, intraoral scan, CAD/CAM

## Abstract

Objective: To assess and compare the accuracy of the marginal and internal fit of lithium disilicate crowns and onlays fabricated by conventional and digital methods. Sources: An electronic search was carried out on MEDLINE, Embase, Web of Science and Cochrane Library between 2010 and 2021. Study selection: Seventeen studies published between 2014 & 2021 were included, of which thirteen were in vitro laboratory-based studies; three were in vivo clinical studies and one randomised controlled trial. Data: Twelve studies focused on the marginal fit, five focused on the marginal and internal fit. Five studies found that the marginal and internal fit of crowns were more accurate using digital techniques. Five studies noted that there was no difference using either technique and two noted that conventional methods had a more accurate marginal fit. Conclusion: Digital techniques were comparable to conventional methods in terms of accuracy although there was insufficient evidence to indicate that one technique was more accurate than the other with respect to Lithium Disilicate restorations. Clinical significance: Digital impressions are reliable and viable alternatives for clinicians compared to conventional impression techniques when restoring teeth with lithium disilicate restorations.

## 1. Introduction

Dental impression materials have been used since the 1800s [1] as a reliable and accurate way of transferring the shape of teeth from the mouth to the dental laboratory and as such are an invaluable resource for both clinicians and dental technicians allowing teeth to be replicated outside of the mouth. Prosthetic and restorative work such as the fabrication of dentures, crowns and bridges are all heavily dependent on accurate and stable impressions for the technician to work with [2]. As technology has developed, digital scanners have been used in many different sectors from producing surgical guides to digitally designing components and machine parts [3]. Dentistry is one area where the introduction of scanning has changed the way treatment can be undertaken. Dental scanners can record teeth, gingivae and implants without the use of impression materials to develop highly accurate records for the technician when making a restoration to ensure the marginal and internal fit to the underlying tooth is as accurate as possible which directly affects the longevity of the restoration and tooth [4]. No one technique is without its drawbacks; several challenges can occur when taking conventional impressions such as distortions, casting errors and damage to the casts during transportation [5]. Intra-oral scanners (IOS) can circumvent some of these issues and specific areas can be rescanned where required to improve accuracy without the need to retake an impression. However, when two margins are very close together, whether they are supra or sub-gingival, achieving accuracy can be challenging. This can significantly reduce the amount of chairside time clinicians need with patients, which results in a more efficient and effective workstream. The main drawbacks of scanners are the high costs involved and recording deep subgingival margins of preparations. 

The rapidly expanding field of digital dentistry offers a reliable alternative to accurately recording tooth shapes following preparation and enables treatment planning to be performed. Information can be sent instantaneously and smoothly via an electronic link to the laboratory compared to traditional methods that have been used for many years. Direct restorations, such as fillings, are placed immediately by dentists but more extensive indirect restorations such as crowns, bridges and onlays utilise different materials, such as lithium disilicate, and require the skills of a suitably trained technician. Popularity for IOS within dentistry has grown for several reasons including: reduced costs for impression materials, the ability to easily repeat scans for areas that have not been captured accurately at the first attempt and an improved experience for patients who suffer with particularly sensitive gag reflexes, requiring less chair time [6]. Potential problems with traditional, fragile gypsum models can be avoided by fabricating three dimensional (3D) printed models from digital scans. A Standard Tessellation Language (STL) file is generated when a scan is taken, and this can be used to 3D print models using CAD/CAM software. 3D models can be made by a stereolithography additive technique (SLA) or by a subtractive method involving milling [7]. The subtractive method poses an issue if there are undercuts or complex shapes that need to be fabricated as the axes of the machine is limited [8]. Errors can also be introduced due to the diameters of the burs used to mill the model [9]. Models can be indirectly scanned and used in the CAD/CAM process, but errors can occur due to the degradation in the impression material and gypsum models [10]. Direct digitalisation is also possible by scanning an impression instead of fabricating a model and subsequently scanning the model [11]. 

With the increased interest in aesthetic materials for both anterior and posterior restorations, the development of technology for the introduction of new materials in the past 16 years has led to the introduction of lithium disilicate as one option. IPS e.max (Ivoclar Vivadent, Schaan, Liechtenstein) was introduced in 2005 and can be used in either a lost wax hot press technique (IPS e.max Press) or designed and milled by a CAD/CAM (IPS e.max CAD) digital technique [12]. 

The accuracy of how well a restoration fits onto a preparation can have a significant effect on the success and survival rate of that restoration and therefore knowledge of the potential inaccuracies or ‘marginal gap’ is important [13]. A marginal gap is defined as the vertical distance from the cervical margin of a restoration to the preparation margin of the tooth [14]. Christensen [15] advised that marginal gaps should measure no more than 120 μm. An alternative study demonstrates that a gap of 100 μm is clinically acceptable [16]. The long-term success of a restoration is highly influenced by both the marginal and internal fit as a poor fit will subsequently lead to secondary caries [17] and marginal inflammation [18]. An internal gap is defined as “the perpendicular distance from the axial wall of the tooth to the internal surface of the restoration” [19]. Now that workflows are changing to become fully digital, the restorations fabricated must be of at least the same accuracy marginally and internally compared to conventional restorations. Systematic reviews have been published previously looking into the marginal and internal fit of restorations made from CAD/CAM technology vs. conventional methods, but none so far have been published solely on indirect lithium disilicate restorations. This systematic review aims to summarise the literature published from studies investigating the marginal and internal fit of lithium disilicate crowns and onlays fabricated with CAD/CAM digital techniques compared with traditional impression methods. 

One should be reminded that marginal and internal fit shall also be considered under function. It has in fact been reported that the marginal gap can increase during cyclic fatigue and depends on preparation design and material [20].

## 2. Materials and Methods

The scoping review was performed in accordance with the Preferred Reporting Items for Systematic Reviews and Meta-Analyses (PRISMA) protocols [21]. 

Null hypothesis: There is no difference in the marginal or internal fit of lithium disilicate crowns and onlays when using digital or conventional techniques. 

### 2.1. Protocol Development

A focused question was formed about the marginal fit accuracy of lithium disilicate crowns and onlays fabricated by digital and conventional methods. A PICO was established to help formulate the search strategy and for development of the inclusion and exclusion criteria. 

P: Lithium disilicate crowns and onlays

I: CAD/CAM restorations

C: Conventional restorations

O: Marginal and Internal fit

### 2.2. Search Strategy

A detailed electronic search was carried out on several databases including MEDLINE, Embase, Web of Science and the Cochrane library. The search used a combination of MeSH terms and focused keywords. The search was carried out from 2010 to 2021, the year the first study investigating CAD/CAM fabricated lithium disilicate crowns and onlays were available. The inclusion and exclusion criteria are summarised in Table 1.

The search strategy was constructed using a combination and variation of the terms CAD/CAM, digital impression, lithium disilicate, marginal adaptation, accuracy, trueness and precision. Time limits were applied, and the results were limited to English. The databases chosen were Medline, Embase, Web of Science and Cochrane. MeSH terms were included.

### 2.3. Study Selection and Data Extraction

Screening of potential articles for analysis was undertaken using in order: (i) title and abstracts; (ii) reading the full text and (iii) additional studies from bibliographies.

### 2.4. Quality of Evidence Assessment

A quality of evidence assessment was carried out using the Cochrane collaboration tool for in vivo studies and an adapted methodological index for nonrandomized studies (MINORS) [22] was used for comparable in vitro studies. The studies in the MINORS table were given a score from 0 to 2 where 0 = the category was not reported, 1 = the content was reported but inadequately and 2 = the information was reported adequately. A maximum possible score was out of 20. A high risk of bias was deemed to be a score under 10, an unclear risk was deemed to be under 16 and a low risk was any score that was 16 and above. These scores were influenced by the paper that discusses the original MINORS scale [22]. 

Figure 1 illustrates the process adopted for selecting relevant studies to review. 180 articles were excluded following review of their titles and abstracts as they did not fit the inclusion criteria. Further exclusions were considered due to: (i) flawed methodology, (ii) lack of related data, (iii) no results table and (iv) no standard deviations reported.

### 2.5. Analysis

Scanners were divided into two categories, namely “intraoral scanner” or “laboratory scanner”. Due to the varying number of different sites measurements reported by the different studies an average of the mesial and distal values were taken and termed the proximal marginal fit value. Statistical analysis and meta-analysis were carried out using STATA software (StrataCorp. College Station, Texas, USA 17.0/20 April 2021). 

Table 2 summarises the authors and titles of the papers that were considered for the analysis. Each paper was allocated a reference number, which is then used as an indicator throughout the ensuing text.

### 2.6. Meta-Analysis

Available data was summarised from each paper by means (SD) where available for CAD/CAM and controls. Data was not available for all papers, and more than one result was often reported in others. The results were summarised into: (i) overall marginal fit across all aspects of the restoration; (ii) inter-proximal marginal fit, an average of reported mesial and distal results and iii) internal fit.

For purposes of meta-analyses of available data was summarised from each paper by means (SD) for CAD/CAM and controls. Some papers provided more then one result (site specific, overall, scanner type); some studies failed to provide sufficient detail.

## 3. Results 

### 3.1. Study Characteristics

The characteristics of the papers were summarised (See Table 3). Maintaining the restoration on the prepared dye was undertaken by a variety of methods whilst measuring the marginal gap: finger pressure, silicone pressure and try in paste. Only one study [25] cemented the restoration on to the dye using zinc phosphate cement. Custom devices were used in three studies [23,26,32] that applied occlusal force to the restoration. Marginal fit was investigated by the majority of the studies except for (four) which looked at internal fit as well and one that used internal volume. The measurement techniques used varied from using stereomicroscopes, a replica technique using silicone and micro-CT imaging. 

### 3.2. Outcome of Studies 

The findings from the papers are summarised below in Table 4 and Table 5. The characteristics of included studies are summarized in Table 6.

Figure 2 shows the full results available from all studies. The heterogeneity (87.2%; *p* < 0.001) is considerable and places limits on the generalisability of the pooled result. However, the pooled estimate 0.07(95%CI−0.53, 0.38) suggests that there is no significant advantage to CAD/CAM compared with Control. To investigate the heterogeneity, a comparison was made between Intraoral scanners (pooled estimate −0.05(95%CI −0.40, 0.50) and lab-based scanners (−0.07(95%CI −0.53, 0.48) but did not reduce the heterogeneity (I2 = 86.2% and 92.6%, respectively). Further comparison between subgroups of potential confounders was handicapped by the number of different scanners and methods employed. Studies by Elrashid et al. (2019) and Anadioti et al. a & b (2014) contributed to the heterogeneity.

Figure 3 suggests that there is no evidence of publication bias but clearly displays over-dispersion caused by the heterogeneity.

### 3.3. Risk of Bias in Studies 

All the in vitro studies had a low risk of bias in terms of the six items used in the quality assessment tool. There was a similar low risk of bias shown for the in vivo studies summarised in Figure 4 by the Cochrane Collaboration tool. Table 7 illustrates the MINORS score for all chosen studies. Table 8 shows details of the risk of bias calculated for each study included in the analysis.

The results presented in Figure 5 relate to the marginal fit of crowns which were not split into an intra-oral or laboratory scanner category. Figure 5 shows that all the point estimates for the interproximal fit support CAD/CAM scanners. All the studies had wide confidence intervals indicating that the estimate has a lower degree of precision. The pooled estimate of the main effect (−0.91(95%CI −1.39, −0.43)) emphasises this difference. However, the heterogeneity (I^2^ = 76%) shows that there remain unexplained sources of variation and the pooled estimate needs to be treated with caution.

Figure 6 shows that, with one exception, all the point estimates for the internal fit support Control scanners. The pooled estimate of the main effect (−0.89 (95%CI −0.31, 2.10)) suggests that this difference is not statistically significant and the heterogeneity ((I^2^ = 92.7%) 76%) shows that there remain unexplained sources of variation and the evidence of any difference is equivocal. Figure 6 illustrates a high degree of heterogeneity with an I^2^ value of 92.7%. This is also represented by wide confidence intervals present for the 7 data points. The majority of the results were significantly in favour of conventional techniques as they are skewed to the right-hand side of the line of no effect (‘0′). 

### 3.4. Synthesis of the Meta-Analysis

A high degree of heterogeneity was present in all forest plots with I^2^ values between 76% to 92.7%. Broadly, all confidence intervals are wide for all the reported results indicating that the estimate has a lower degree of precision. The internal fit results favoured conventional methods whereas the marginal fit results for crowns favoured CAD/CAM methods. There were several outlier results reported by Lee et al. [36] and Elrashid et al. [32]. 

## 4. Discussion 

### 4.1. Summary of Evidence

The results of this review were equivocal, there was insufficient evidence to reject the null hypothesis. This was largely due to the high level of heterogeneity present in the studies and the quality of the evidence. This high degree of heterogeneity is multi-factorial and is also present in past reviews. These results have been similarly found in recent systematic reviews that have been carried out comparing the accuracy of 3D digital techniques compared to conventional methods [39,40,41]. One of the possible reasons why the heterogeneity may have been present could have been due to the type of scanner used in the study or due to the type of study, e.g., an in vivo or in vitro study. 

### 4.2. Measurement Technique 

There were a variety of results reported across all the studies; the lowest marginal gap recorded was from Elrashid et al. [33]. A custom-made holder was used with a pin to lock the restoration in place on the dye when measuring the marginal gap, which ensured a very close fit to the corresponding dye. This variation in methodology could have influenced the result as many of the other studies used silicone or finger pressure to seat the restoration on the dye. The method of measuring the marginal or internal gap on a restoration is critical as some techniques are more accurate and reliable than others. Some of the studies [22,25,30,37] utilised µ-CT which employs high level of accuracy and is a non-destructive method which can be done from any angle and can give 2D and 3D measurements [42]. This is not a possible option for clinical in vivo studies, where it is more challenging to perform gap measurements due to saliva and patient compliance. The replica technique was used by many of the studies and has been a well-documented method used for many years [43]. However, only a certain number of sections can be made, and defects can be present in the silicone leading to inaccuracies. This current lack of consensus when measuring the marginal or internal gap for indirect restorations could be a result of several different methods of measuring used in studies [44].

### 4.3. Internal Fit 

There were only a few studies investigating the internal fit of crowns and onlays compared to the marginal fit. It was also challenging to accurately measure this parameter as 2-Dimensional measurements are of limited use since the internal surfaces of a prepared tooth vary and are not uniform [45]. The smallest internal fit measurements across the studies were found by Kim et al. [26] that used µ-CT to ascertain a 3-D interval volume measurement. This was also carried out by Revilla-Leon et al. [31] who found similarly low results. These results are comparable to the study by Alfaro et al. [46] that solely investigated the internal fit of lithium disilicate crowns using µ-CT. The overall internal results found by Zeltner et al. [28] and Lee, Son and Lee [37] were much higher as these were clinical in vivo studies where µ-CT was not possible, and the replica technique was used in both these studies. Therefore, the results are almost impossible to compare as µ-CT would give a far more accurate result. 

### 4.4. Onlays 

Onlay preparations can vary in design and if there are any intricate preparation features these can be difficult for milling machines to accurately recreate the shapes; it has been suggested that a small 0.6mm bur should be used when fabricating onlays with complex geometry [26]. There were no clinical studies carried out using onlay restorations which makes it difficult to reach a conclusion using two laboratory-based studies alone. Clinical studies using onlay preparations could give differing results to crowns as the preparation margins of onlays are generally supragingival and easier to record. Onlay restorations are becoming increasingly more popular and common within general practice due to the conservative nature of the preparation which is in keeping with the trend of minimally invasive dentistry [47].

### 4.5. CAD/CAM and Milling Machines

It has been found that CAD/CAM fabricated restorations commonly have small marginal gaps and larger occlusal gaps [48]. Additional space is required around cusps to compensate for the size of the tools used to mill the material [49] which can result in larger internal gaps. Five-axis milling machines have been shown to grind in deeper and narrower areas which could result in a restoration with a better marginal fit and greater accuracy [23,50]. Overall, many of the marginal discrepancy measurements reported by the studies were below 120 µm which McLean and Von Fraunhofer [51] suggested as a clinically acceptable limit where a restoration would be successful. 

### 4.6. Limitations 

There were very few in vivo studies included [27,31,34,36], of which, only one was carried out as a randomised controlled trial with the lowest risk of bias [27]. The results from the studies using models rather than real teeth will inherently produce differing results. Therefore, this needs to be considered when comparing the results of the in vitro and in vivo studies. Statistical analysis could not be carried out on the individual scanner type to determine if there was any significant difference and an average interproximal value had to be used. The sample size used in these studies was small implicating that the findings might be different if the investigations were to be repeated on a larger scale. 

The heterogeneity in all these analyses prohibits a precise estimate and conclusion. The evidence points towards a superiority of CAD/CAM when used in for interproximal measurement, which is corroborated by the results in Table 4 reported in the first part of the results, but the heterogeneity results show that the evidence is unclear.

### 4.7. Future Developments

Future studies using a similar protocol in terms of preparation of the teeth, the type of scanner, the milling machines employed and measurement techniques, must be completed to get more realistic results, which can be compared, enabling a more confident prediction whether there is a significant difference in using digital or conventional techniques. All these studies investigated single unit crowns, but the results could be different when applied to bigger restorative cases involving long span bridges or multiple crowns in an arch. Currently, there are too many different variables within different studies to be able to confidently advise clinicians when choosing to use either digital or conventional methods for impression taking in clinical scenarios in general practice.

## 5. Conclusions

Digital technology is a viable alternative to conventional impressions for fabrication of lithium disilicate crowns and onlays. The majority of results demonstrated that the restorations, digital or conventional, would have a high success rate for patients. However, there is insufficient evidence to suggest that digital methods can replace the use of conventional impression materials. A multitude of variables exist such as types of scanners available on the market, milling machines and 3-D printed models which would make it more challenging to arrive at a conclusive answer as to which technique is superior, as any of the variables could affect the outcomes. Further clinical in vivo studies of higher quality, e.g., randomised controlled trials are needed as they will provide realistic scenarios and meaningful information which might not be apparent in laboratory-based studies. Further guidance and clarity are needed on measurement techniques for marginal and internal gaps as the disparity in the methods used in studies means the information cannot be easily compared or used in a clinical scenario.

## Figures and Tables

**Figure 1 dentistry-10-00236-f001:**
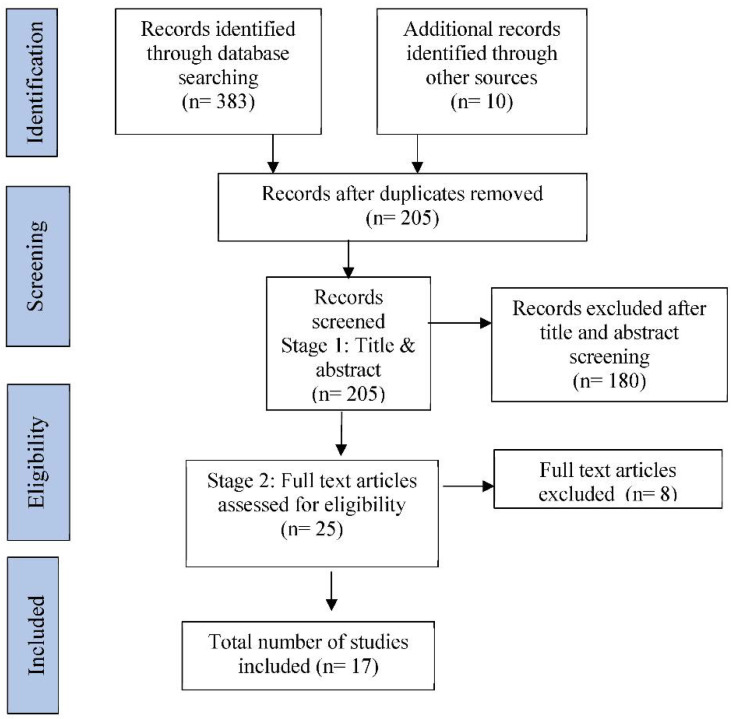
PRISMA diagram.

**Figure 2 dentistry-10-00236-f002:**
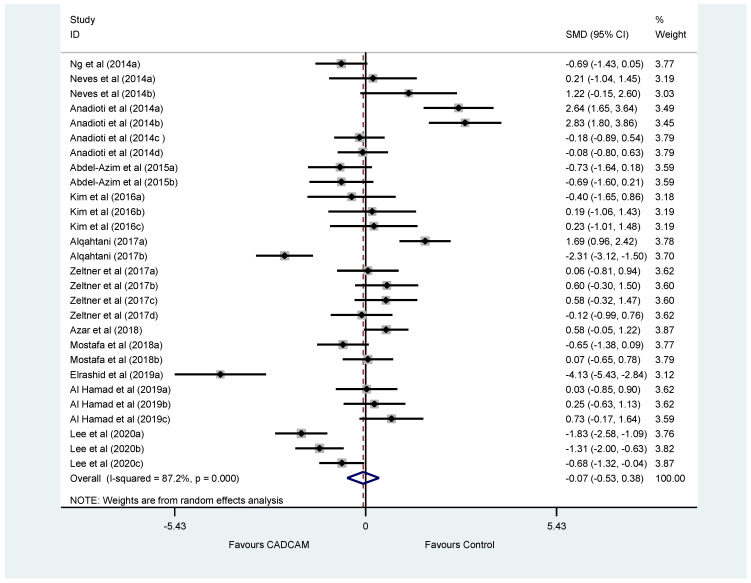
Forest plot showing standardised mean difference (SMD) comparing the overall marginal fit of crowns by CAD/CAM with controls included studies investigating the marginal fit of crowns using an intra-oral or laboratory scanners. [Ng et al., 2014; Neves et al., 2014; Anadioti et al., 2014; Abdel-Azim et al., 2015; Kim et al., 2016; Alqahtani 2017; Zeltner et al; 2017; Azar et al., 2018; Mostafa et al; 2018; Elrashid et al; 2019; Al Hamad et al; 2019; Lee et al; 2020].

**Figure 3 dentistry-10-00236-f003:**
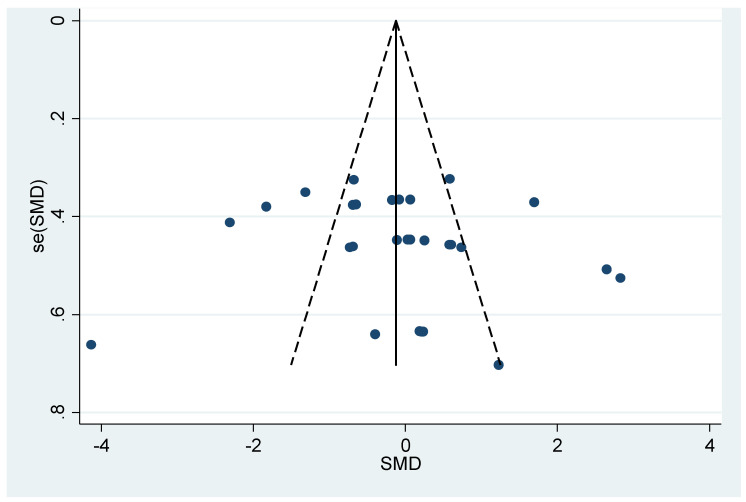
Funnel Plot of Overall Marginal Fit.

**Figure 4 dentistry-10-00236-f004:**
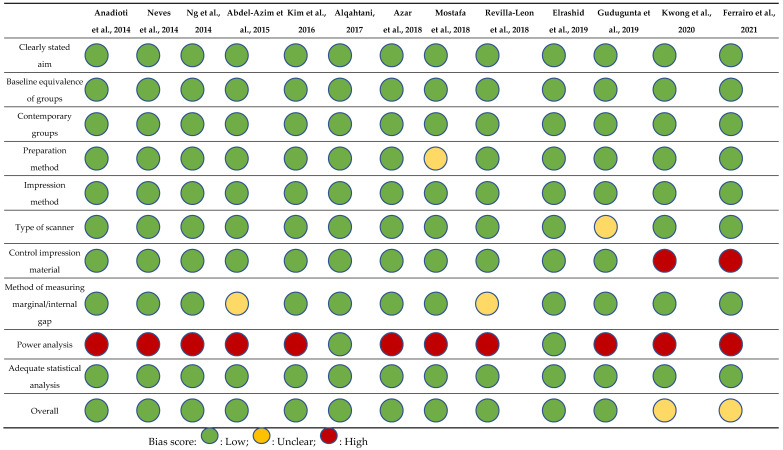
Risk of bias summary graph of included studies.

**Figure 5 dentistry-10-00236-f005:**
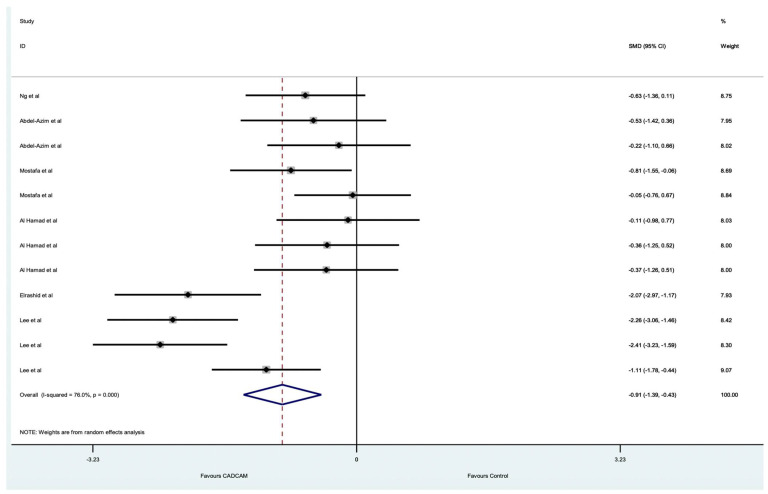
Forest plot comparing the included studies investigating the inter-proximal marginal fit of crowns.

**Figure 6 dentistry-10-00236-f006:**
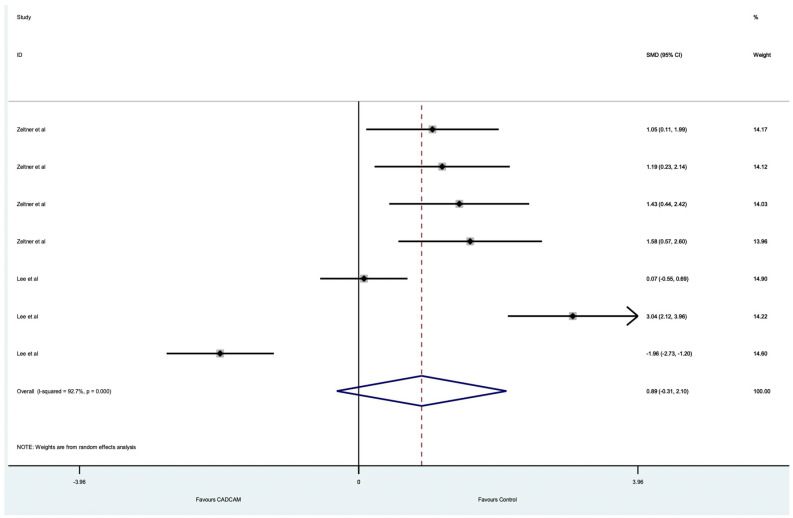
Forest plot of included studies investigating the internal fit of crowns.

**Table 1 dentistry-10-00236-t001:** Inclusion/Exclusion criteria for the literature review.

Inclusion Criteria	Exclusion Criteria
Articles published in English Title and abstract relevant to question Dates from 2010–2021 Peer reviewed papers Studies discussing lithium disilicate crowns and onlays only Studies that investigated marginal fit and internal fit. Comparative studies which are in vivo or in vitro	Articles unrelated to the PICO questions studies Papers discussing inlays/ implants/ partial dentures Studies that investigate any material that is not lithium disilicate Papers looking at success rate/ survival rate/ time efficiency/ fracture strength No clear mention of CAD/CAM system trade name No clear mention of measurement technique Qualitative assessment of marginal fit

**Table 2 dentistry-10-00236-t002:** Studies included in the analysis.

Author	Title
Anadioti et al., 2014 [12]	3D and 2D Marginal Fit of Pressed and CAD/CAM Lithium Disilicate Crowns Made from Digital and Conventional Impressions
Neves et al., 2014 [23]	Micro-computed tomography evaluation of marginal fit of lithium disilicate crowns fabricated by using chairside CAD/CAM systems or the heat-pressing technique
Ng, Ruse and Wyatt, 2014 [24]	A comparison of the marginal fit of crowns fabricated with digital and conventional methods
Abdel-Azim et al., 2015 [25]	Comparison of the marginal fit of lithium disilicate crowns fabricated with CAD/CAM technology by using conventional impressions and two intraoral digital scanners
Kim et al., 2016 [26]	Fit of lithium disilicate crowns fabricated from conventional and digital impressions assessed with micro-CT.
Alqahtani, 2017 [27]	Marginal fit of all-ceramic crowns fabricated using two extraoral CAD/CAM systems in comparisonwith the conventional technique.
Zeltner et al., 2017 [28]	Randomized controlled within-subject evaluation of digital and conventional workflows for the fabrication of lithium disilicate single crowns. Part III: marginal and internal fit
Azar et al., 2018 [29]	The marginal fit of lithium disilicate crowns: Press vs. CAD/CAM
Mostafa et al., 2018 [30]	Marginal Fit of Lithium Disilicate Crowns Fabricated using Conventional and Digital Methodology: A Three-DimensionalAnalysis.
Revilla-Leon et al., 2018 [31]	Marginal and Internal Gap of Handmade, Milled and 3D Printed Additive Manufactured Patterns for Pressed Lithium Disilicate Onlay Restorations.
Al Hamad et al., 2019 [32]	Comparison of the Fit of Lithium Disilicate Crowns made from Conventional, Digital, or Conventional/DigitalTechniques
Elrashid et al., 2019 [33]	Stereomicroscopic Evaluation of Marginal Fit of E.Max Press and E.Max Computer-Aided Design and Computer-Assisted Manufacturing Lithium Disilicate Ceramic Crowns: An In vitro Study
Gudugunta et al., 2019 [34]	The marginal discrepancy of lithium disilicate Onlays: Computer-aided design versus press
Ahn et al., 2020 [35]	Clinical evaluation of the fit of lithium disilicate crowns fabricated with three different CAD-CAM systems.
Kwong and Dudley, 2020 [36]	A comparison of the marginal gaps of lithium disilicate crowns fabricated by two different intraoral scanners
Lee, Son and Lee, 2020 [37]	Marginal and Internal Fit of Ceramic Restorations Fabricated Using Digital Scanning and Conventional Impressions: A Clinical Study
Ferrairo et al., 2021 [38]	Comparison of marginal adaptation and internal fit of monolithic lithium disilicate crowns produced by 4 different CAD/CAM systems

**Table 3 dentistry-10-00236-t003:** Illustrates each Study Characteristics.

In vitro studies	13 [12,23,24,25,26,27,29,30,31,33,34,36,38]
In vivo studies	4 [28,32,35,37]
Onlays	2 [31,34]
Crowns	15 [12,23,24,25,26,27,28,29,30,32,33,35,36,37,38]
PVS control used	14 [12,23,24,25,26,27,28,29,30,31,32,33,34,37]
Digital only	3 [35,36,38]

**Table 4 dentistry-10-00236-t004:** Illustrates the Conclusions Drawn from the included studies.

Digital techniques resulted in a more accurate marginal fit of crowns	5 [24,27,30,33,37]
Conventional techniques resulted in a more accurate internal fit of crowns	1 [28]
No clear difference in accuracy of internal fit in either method	2 [26,37]
Unable to compare internal fit of onlay to any other study	1 [31]
No difference in marginal fit with either method	5 [23,25,26,28,32]
Conventional techniques resulted in a more accurate marginal fit of crowns	2 [12,29]
Digital only methods- no difference between digital methods	2 [35,36]
Digital only methods- significant difference between digital methods	1 [38]
Conventional techniques resulted in a more accurate marginal fit of onlays	1 [31]
Digital techniques resulted in a more accurate marginal fit of onlays	1 [34]

**Table 5 dentistry-10-00236-t005:** Reported marginal and internal gaps in the included studies.

Largest marginal gap for crowns	207.8 μm [26]
Smallest marginal gap for crowns	26.8 μm [33]
Largest internal gap for crowns	285.2 μm [28]
Smallest internal gap for crowns	80 μm [37]
Smallest marginal gap for onlays	41.5 μm [34]
Largest marginal gap for onlays	86.5 μm [31]
Smallest internal gap for onlays	92 μm [31]
Largest internal gap for onlays	96 μm. [31]

**Table 6 dentistry-10-00236-t006:** Illustrates Characteristics of the included studies.

Author	Type of Study	Type of Scanner	Comparison	Measurement Technique	Outcome of Study	Outcomes
Anadioti et al., 2014 [12]	In vitro study	E4D scanner	PVS impressions	Triple scan protocol and 2D measurements measured at 4 points	Marginal fit	PVS impression and press fabrication technique the most accurate for 3D and 2D marginal fits.
Neves et al., 2014 [23]	In vitro study	CEREC 3D Bluecam and E4D scanner	PVS impressions	Scanned with μ-CT. For every image 2 measurements were taken for horizontal and vertical fit at 400× magnification. The crowns were fixed to the die with silicone material.	Marginal fit	Lithium disilicate crowns fabricated using CEREC 3D Bluecam scanner CAD/CAM system or the heat press technique more accurate than crowns fabricated using an E4D Laser scanner system.
Ng, Ruse and Wyatt, 2014 [24]	In vitro study	Lava COS scanning unit	PVS impressions	Circumferential marginal gap measurements takenat 8 measurement locations. photographs at 40× with a Die was held in place with a measurement alignment chuck	Marginal fit	The fully digital fabrication method provided better margin fit than the conventional method
Abdel-Azim et al., 2015 [25]	In vitro study	Lava COS scanning unit and iTero	PVS impressions	Stereomicroscope with a microscope camera used at 45× magnification and measured at 4 points. A computer software used.	Marginal fit	Digital and conventional impressions were found to produce crowns with similar marginal accuracy
Kim et al., 2016 [26]	In vitro study	CS3500/ TRIOS/ Ceramill Map400	PVS impressions	Crowns were cemented to replica die with zinc phosphate cement and finger pressure. 2D and 3D measurements were used to measure the marginal gap. Internal volume measured using volume of zinc phosphate cement between lithium disilicate crown and replica die.	Marginal fit and internal volume	Significant differences were found between e.max CAD crowns produced using 2 intra-oral digital impressions.
Alqahtani, 2017 [27]	In vitro study	CEREC omnicam and TRIOS	PVS impressions	A loading device was used to apply an occlusal load of 3 lbs during the measurement. Scanning electron microscope at a magnification of 50× used to measure. Three measurements were taken for each point, and average was recorded.	Marginal fit	All-ceramic crowns, fabricated using the CAD/CAM system, show a marginal accuracy that is acceptable in clinical environ- ments. The TRIOS CAD group displayed the smallest marginal gap.
Zeltner et al., 2017 [28]	Randomised controlled trial	Lava, iTero, CEREC inLab, and CEREC infinident systems	PVS impressions	Replica technique used applying finger pressure. Replica sectioned and measurements taken with light microscopy at 200× magnification using a blinded investigator.	Marginal and internal fit	No significant differences were found between the conventional and digital workflows for the fabrication of monolithic lithium disilicate crowns.
Azar et al., 2018 [29]	In vitro study	CEREC Omnicam	PVS impressions	Measurements taken using an optical microscope at 200× magnification. The measurements were performed on 25 points on the finishing line of each tooth	Marginal fit	Lithium disilicate crowns fabricated with the press technique have measurably smaller marginal gaps compared with those fabricated with CAD/CAM technique.
Mostafa et al., 2018 [30]	In vitro study	Lava C.O.S. scanning unit	PVS impressions	Crowns were seated using finger pressure and wax. 3D volume was measured circumferentially based on a standardized number of slices for the selected region of interest. Vertical marginal gaps were also measured.	Marginal fit	The results suggested that digital impression and CAD/CAM technology is a suitable, better alternative to traditional impression and manufacturing.
Revilla-Leon et al., 2018 [31]	In vitro study	Laboratory based scanner Renishaw DS20.	PVS impressions	Computed tomography (μ-CT) for measurement of marginal and internal gaps.	Marginal and internal fit	All the groups presented less than 100 μm marginal and internal gap. The best marginal and internal fit was still obtained by the conventional handmade procedures.
Al Hamad et al., 2019 [32]	Prospective controlled clinical trial	CEREC Omnicam	PVS impressions	Crowns seated using silicone and finger pressure. Replica technique used and gaps analysed using a stereomicroscope at 30× magnification.	Marginal fit	Ceramic crowns, which were made using all-digital approach or cast digitization by a laboratory or intra-oral scanner had comparable fit to those produced by conventional approach.
Elrashid et al., 2019 [33]	In vitro. study	Laboratory scanner Exocad smart optical 3D-scanner	PVS impressions	Crowns held on to the die using a custom made holder with a pin. Marginal gap measured using a digital microscope at 50× magnification.	Marginal fit	Lithium disilicate all ceramic crowns fabricated by using CAD-CAM techniques showed lesser marginal gap and better marginal fit compared to the conventional technique.
Gudugunta et al., 2019 [34]	In vitro study	CEREC scanner	PVS impressions	The crowns were held on to the dies using try in paste. The marginal gap was measured using a microscope camera at 200× magnification.	Marginal fit	Although there was a statistically significant difference between the two groups, marginal gap of both the groups were in clinically acceptable levels.
Ahn et al., 2020 [35]	Prospective controlled clinical trial	CEREC Bluecam, EZIS PO and TRIOS 3 scanners	Comparison with other intra-oral scanners	Crowns seated using silicone and finger pressure. Silicone replica technique used measured using a microscope at 100× magnification.	Marginal fit	The lithium disilicate crowns of all groups showed clinically acceptable fit.
Kwong and Dudley, 2020 [36]	In vitro study	TRIOS 3 and E4D scanner	Comparison with other intra-oral scanners	Crowns seated using silicone and finger pressure. Digital stereomicroscopy with a camera at a magnification range of 2–18×. Three measurements were taken at each of the 4 locations	Marginal fit	There was no difference in the marginal gaps of CAD/CAM lithium disilicate crowns constructed using two different intra-oral scanners of different generations.
Lee, Son and Lee, 2020 [37]	Prospective controlled clinical trial	EZIS PO, i500, and CS3600 scanners	PVS impressions	Silicone replica technique measured at 60× magnification	Marginal and internal fit	There was a significant difference in the marginal and internal fit of the ceramic crowns fabricated using three intra-oral scanner types and one desktop scanner type
Ferrairo et al., 2021 [38]	In vitro study	CERAMILL, CEREC, EDG and Zirkonzahn scanners	Comparison with other intra-oral scanners	Crowns seated on to dies using silicone. Silicone replica technique used. internal space measured using μ-CT and replica technique images by a film thickness images were captured by means of a stereomicroscope at 50× magnification.	Marginal and internal fit	The 4 systems are capable to produce restorations adapted within clinically appropriate levels.

**Table 7 dentistry-10-00236-t007:** MINORS table of included studies.

	Anadioti et al., 2014 [12]	Neves et al., 2014 [23]	Ng et al., 2014 [24]	Abdel-Azim et al., 2015 [25]	Kim et al., 2016 [26]	Alqahtani, 2017 [27]	Azar et al., 2018 [29]	Mostafa et al., 2018 [30]	Revilla-Leon et al., 2018 [31]	Elrashid et al., 2019 [33]	Gudugunta et al., 2019 [34]	Kwong et al., 2020 [36]	Ferrairo et al., 2021 [37]
Clearly stated aim	2	2	2	2	2	2	2	2	2	2	2	2	2
Baseline equivalence of groups	2	2	2	2	2	2	2	2	2	2	2	2	2
Contemporary groups	2	2	2	2	2	2	2	2	2	2	2	2	2
Preparation method	2	2	2	2	2	2	2	2	2	2	2	2	2
Impression method	2	2	2	2	2	2	2	2	2	2	2	2	2
Type of scanner	2	2	2	2	2	2	2	2	2	2	1	2	2
Control impression material	1	1	1	1	1	1	2	1	1	1	1	0	1
Method of measuring marginal/internal gap	2	2	2	2	2	2	2	2	1	2	2	1	2
Power analysis	0	0	0	0	0	2	0	0	0	2	0	0	0
Adequate statistical analysis	2	2	1	1	2	1	2	1	1	1	1	1	1
Overall	17	17	16	16	16	18	18	16	15	18	15	14	16

**Table 8 dentistry-10-00236-t008:** Risk of bias table of included studies.

	Zeltner et al., 2017 [28]	Al Hamad et al., 2019 [32]	Ahn et al., 2020 [35]	Lee, Son and Lee, 2020 [37]
Type of study	Randomised controlled trial	Prospective controlled clinical trial	Prospective controlled clinical trial	Prospective controlled clinical trial
Adequate sequence generation	Yes—low risk	Unclear	Unclear	Unclear
Remarks	If 2 or more teeth were available on one patient then just 1 tooth was selected by throwing a die.	One patient with a peg-shaped maxillary left lateral incisor was selected for the study.	40 participants were used (10 men/ 30 women). No mention of any sequence generation.	20 participants were used who required a ceramic crown. A power analysis was used.
Allocation concealment	Yes- low risk	Unclear	Unclear	Unclear
Remarks	The sequence of the crown assessment was randomly allocated to a computer generated list.	No mention of how the different methods were allocated.	Each preparation was scanned using 3 different scanners. No mention of any concealment.	No mention of how the different methods were allocated.
Blinding of outcome assessment	Yes- low risk	Unclear	Unclear	Yes—low risk
Remarks	One blinded investigator measured all specimens. 3 calibrated clinicians carried out the procedures	Tooth preparations were carried out by a prosthodontist (KQ) and a different prosthodontist (BR) was in charge of the design and milling.	Tooth preparations were carried by an experienced prosthodontist (JBH). The internal fit was evaluated by an examiner (JJA)	The dentist was blinded to the information about the crown. All intraoral processes were prepared by one skilled dentist. A prosthodontist evaluated the quality of the crowns based on the clinician satisfaction score.
Incomplete outcome data addressed	Yes	Yes	Yes	Yes
Remarks	No incomplete data	No incomplete data	Seven drop-outs	No incomplete data
Free of selective reporting	Yes	Yes	Yes	Yes
Remarks	The study protocol is available and the primary/secondary outcomes have been reported on.	The study protocol is available and the primary/secondary outcomes have been reported on.	The study protocol is available and the primary/secondary outcomes have been reported on.	The study protocol is available and the primary/secondary outcomes have been reported on.
Free of other sources of bias	No- declaration made about support and funding.	Yes	No- declaration made about support and funding.	No- declaration made about support and funding..
Remarks	The study was supported by the Clinic of Fixed and Removable Prosthodontics and Dental Material Science, Center of Dental Medicine, University of Zurich, Switzerland, and by a research grant from Institut Straumann AG, Basel, Switzerland.	None declared	Supported by Industrial Strategic Technology Development Program. Funded By the Ministry of Trade, Industry & Energy (MOTIE, Korea).	Supported by the Ministry of Trade, Industry & Energy (MOTIE, Korea) under the Industrial Technology Innovation Program
Overall risk of bias	Low	Medium	Medium	Medium

## Data Availability

All data used in this study was available on web sites searched including MEDLINE, Embase, Web of Science and the Cochrane library.

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
