# Peer review of "A Scoping Review of Marginal and Internal Fit Accuracy of Lithium Disilicate Restorations"

_dentistry, 2022, doi:10.3390/dj10120236_

Round 1

Reviewer 1 Report

Dear Authors,

generally, the write-up of the paper is good. Here follow some suggestion in order to improve the manuscript.

Here follow some suggestions:

Due to the huge heterogeneity of the included studies the reviewer thinks that maybe a Scoping review could have been a better study. By the way systematic could be still a valid tool.

Line 42:

Please add also the concept that when two teeth are very close, a correct recording is very challenging, despite is subgingival or not.

Line 79 till the end of the Introduction:

The reviewer thinks that mentioning consequences of a poor fit during function should be mentioned. And should also be mentioned in the discussion/limitation. A sentence like the following could help the reader to understand the importance of a cyclic fatigue study:

“It has also to be reminded that marginal and internal fit shall be also considered under function. It has in fact been reported that marginal gap can increase during cyclic fatigue and depends on preparation design and material” The authors could cite: https://doi.org/10.1111/jerd.12837 to support this sentence

Line 80:

A poor fit of an indirect restoration should not be considered the trigger of a periodontal disease. The reviewer thinks that “marginal inflammation” could be a more proper term.

Line 113:

Exclusion criteria: 

“Papers looking at success rate/ survival rate/ time efficiency/ fracture strength”

and

“Studies that investigate any material that is not  lithium disilicate”
are redundant, they are automatically excluded by the inclusion criteria

Line 115:

Please specify that title and abstract were retrieved at first.

then followed full-texts taken from the bibliographies

Line 119:

In this review the majority of the included papers are in-vitro papers.

The Risk-of-bias tool MINORS is not meant to be applied to in-vitro studies but to in-vivo non randomized studies. Please explain and/or support this decision.

Line 160:

Figure 3 is mentioned. The reviewer is not able to find Figure 3.

The reviewer is not able to find Figure 2, its caption and its citation in the manuscript.

Most of the tables are not cited in the text.

Author Response

Dear Authors,

generally, the write-up of the paper is good. Here follow some suggestion in order to improve the manuscript.

Here follow some suggestions:

Due to the huge heterogeneity of the included studies the reviewer thinks that maybe a Scoping review could have been a better study. By the way systematic could be still a valid tool. The title has been changed to reflect Reviewer 1 's suggestion

Line 42:

Please add also the concept that when two teeth are very close, a correct recording is very challenging, despite is subgingival or not. This issue has been addressed as suggested by the reviewer.

Line 79 till the end of the Introduction:

The reviewer thinks that mentioning consequences of a poor fit during function should be mentioned. And should also be mentioned in the discussion/limitation. A sentence like the following could help the reader to understand the importance of a cyclic fatigue study:

“It has also to be reminded that marginal and internal fit shall be also considered under function. It has in fact been reported that marginal gap can increase during cyclic fatigue and depends on preparation design and material” The authors could cite: https://doi.org/10.1111/jerd.12837 to support this sentence. We would like to thank the reviewer for their very appropriate suggestion, including reference; this has been incorporated into the manuscript.

Line 80:

A poor fit of an indirect restoration should not be considered the trigger of a periodontal disease. The reviewer thinks that “marginal inflammation” could be a more proper term. This has been changed following the suggestion by reviewer 1

Line 113:

Exclusion criteria: 

“Papers looking at success rate/ survival rate/ time efficiency/ fracture strength”

and

“Studies that investigate any material that is not  lithium disilicate”
are redundant, they are automatically excluded by the inclusion criteria. It was felt necessary to include ‘studies that investigate any material that is not lithium disilicate’ in the exclusion criteria so as to ensure no ambiguity for the reader. This has therefore not been changed.

Line 115:

Please specify that title and abstract were retrieved at first.

then followed full-texts taken from the bibliographies. This has been changed to reflect the suggestions by reviewer 1

Line 119:

In this review the majority of the included papers are in-vitro papers.

The Risk-of-bias tool MINORS is not meant to be applied to in-vitro studies but to in-vivo non randomized studies. Please explain and/or support this decision. Slim et al. suggests that MINORS can be used for both in-vivo and in-vitro studies, which we have followed, and therefore we have not altered this point.

Line 160:

Figure 3 is mentioned. The reviewer is not able to find Figure 3. The numbering of all figures has been updated to correct this.

The reviewer is not able to find Figure 2, its caption and its citation in the manuscript. All figures have now been correctly numbered.

Most of the tables are not cited in the text. This has been corrected

Reviewer 2 Report

Dear authors, 

Along with Inclusion/Exclusion criteria for the literature review please add the search strategy for each database

Why were the (n= 180) records excluded after title and abstract screening? And full text articles excluded (n= 8)?

In the prisma is unclear why 17 studies remained

Table 2 is unnecessary

In table 3 data should be presented in detail, not just numbers in []

The Forrest plot in Figure 4 ,5 and 6 should be better explained in the manuscript

The Summary of evidence should be more detailed.

How can you explain the lack of consensus when measuring the marginal or internal gap for indirect restorations?

To discuss about the internal fit in a review is almost impossible

Results on only 2 studies on onlays are very confusing

This study has many limitations due to the very few in-vivo studies included – therefore should be dealt with caution. How can one rely on them?

What is the originality and novelty this paper brings?

References are not in journal style

Author Response

Dear authors, 

Along with Inclusion/Exclusion criteria for the literature review please add the search strategy for each database. A resume of the search strategy has been added

Why were the (n= 180) records excluded after title and abstract screening? And full text articles excluded (n= 8)? We are grateful to Reviewer 2 for pointing out this lack of clarity and have addressed the issue. The numbering of articles within the PRISMA diagram have been clarified and the explanation as to why 180 and then 8 articles were excluded have been added to the text, thus explaining the PRISMA diagram

In the prisma is unclear why 17 studies remained This has been clarified

Table 2 is unnecessary. It is considered helpful to the reader to be able to see which studies are included at a glance. This allows the authors to save space on Tables 3 & 4.

In table 3 data should be presented in detail, not just numbers in []. The authors felt that by giving each study a number the presentation of the data was clearer to the reader and saved space. This technique for identifying included studies is a well-recognized approach.

The Forrest plot in Figure 4 ,5 and 6 should be better explained in the manuscript. This has been addressed

The Summary of evidence should be more detailed. This has been addressed

How can you explain the lack of consensus when measuring the marginal or internal gap for indirect restorations? The lack of consensus is explained by the varying methodologies, the lack of consistency in reporting results and therefore the high degree of heterogeneity.

To discuss about the internal fit in a review is almost impossible. The internal fit of the restorations was considered to be important to consider/discuss. The fact that a clinician cannot use metal ceramic or Zirconia materials as an onlay, adds to the challenges of this review.

Results on only 2 studies on onlays are very confusing. The authors agree that having only 2 studies looking at onlays adds to the complexity of the study, but feel that to exclude onlays would on balance detract from the overall findings and leave them open to criticism of not including onlays.

This study has many limitations due to the very few in-vivo studies included – therefore should be dealt with caution. How can one rely on them? The authors thank reviewer 2 for this observation and take this point on board. Every effort has been taken to emphasise to the potential readers the inherent limitations of the study as well as the valid points made when interpreting the findings.

What is the originality and novelty this paper brings? The authors feel that this review brings an update on the current thinking about CAD/CAM and its suitability in terms of marginal and internal fit. To our knowledge no previous review has looked specifically at Lithium Disilicate, which given its increased use by dental practitioners, is important for them to read about its suitability as a restorative material.

References are not in journal style. References have been updated to reflect the inclusion of an additional reference; otherwise no changes have been made to the style.

Round 2

Reviewer 1 Report

The authors have amended to all the requests and comments.

Reviewer 2 Report

Congratulations on your work!